# Preoperative evaluation of C2 pedicle screw placement using a deep learning model: Development and validation study

Junhao Bao[1], Wei Wang[2], Yuelin Wu[1,3], Hao Ren[2,3,4], Zhaoquan Liang[5], Qiang Xiao[5], Yeyang Wang[1,3], Fengshi Jing[2,3,4,6], Weibin Cheng[2,3,4,7*], Li Zhang[1,3*]

1 Spine Department, Orthopaedic Center, The Affiliated Guangdong Second Provincial General Hospital of Jinan University, Guangzhou, Guangdong, China, 2 Institute for Healthcare Artificial Intelligence Application, Guangdong Second Provincial General Hospital, Guangzhou, Guangdong, China, 3 Health Science Center, Jinan University, Guangzhou, Guangdong, China, 4 Guangzhou Key Laboratory of Smart Home Ward and Health Sensing, Guangzhou, Guangdong, China, 5 Second School of Clinical Medicine, Southern Medical University, Guangzhou, Guangdong, China, 6 Faculty of Data Science, City University of Macau, Macau SAR, China, 7 School of Data Science, City University of Hong Kong, Hong Kong SAR, China

☯ These authors contributed equally to this work.
* chwb817@gmail.com (WC); lizhang686@163.com (LZ)

## Abstract

### Background

Current preoperative assessment methods for C2 pedicle screw placement face challenges including low consistency, operational complexity, and high skill demands.

### Objective

This study aimed to develop and validate a deep learning model for rapid and accurate assessment of C2 pedicle screw placement feasibility.

### Materials and methods

We developed C2-Net, an automated deep learning pipeline incorporating an image segmentation module for delineating C2 pedicles in CT images and a screw placement probability assessment module. The model's performance was evaluated using 3D-printed manually placed screws as ground truth and compared with surgeons of different experience levels.

### Results

On the test set, C2-Net achieved an accuracy of 89.4%, sensitivity of 90.0%, and specificity of 89.0%. The model demonstrated performance comparable to senior surgeons and numerically superior to junior surgeons, with higher consistency in diagnostic metrics. Attention maps generated by the model provided visual interpretation

**Data availability statement:** The datasets generated and analyzed during the current study are not publicly available due to patient privacy concerns and institutional policies. Requests for access to the data can be submitted to the Medical Ethics Committee of Guangdong Second Provincial General Hospital, which serves as the institutional contact for data access approval (Email: gd2hllwyh@163.com).

**Funding:** This work was supported by the 3D-printing research project of Guangdong Second Provincial General Hospital (3D-A2020006) and the Guangzhou Science and Technology Programme (2024A03J1062, 2024A03J1074, 2023A03J0286, and 2024A03J0927). The funders had no role in study design, data collection and analysis, decision to publish, or preparation of the manuscript.

**Competing interests:** The authors have declared that no competing interests exist.

of the decision-making process. The predicted probabilities demonstrated capability in differentiating structural variations of C2 pedicles.

## Conclusion

C2-Net shows high accuracy and efficiency in assessing C2 pedicle screw placement, outperforming junior surgeons. With its ability to provide rapid, consistent evaluations and visual interpretations, C2-Net demonstrates potential as a valuable assistive tool for clinical decision-making in spinal surgery.

 **Trial Registration:** ChiCTR2500101655

## Introduction

Atlantoaxial pedicle screw fixation is currently the most widely utilized posterior internal fixation technique in the clinic for addressing atlantoaxial instability [1–3]. Among these techniques, pedicle screw fixation of the axis has emerged as the preferred approach for posterior atlantoaxial fixation due to its superior biomechanical properties, minimal complication incidence, and high fusion rates [4,5]. The axis (C2) vertebra, serving as the transitional vertebra between the atlas (C1) and the lower cervical spine, exhibits complex and irregular anatomical morphology and structure. Additionally, the atlas pedicle is in close proximity to the spinal cord and vertebral artery. Consequently, during the placement of atlas pedicle screws, there exists a potential risk of vertebral artery injury, which may lead to intraoperative vertebral artery rupture, compromising blood supply to the vertebrobasilar artery system and thereby posing a threat to patient safety. Moreover, the success of C2 pedicle screw placement often hinges upon the dimensions of the narrowest segment of the patient's pediculoisthmic component (PIC), emphasizing the criticality of preoperative assessment for screw placement feasibility [6].

Our previous study had validated that CT multiplanar reconstruction (MPR) can effectively delineate the trajectory of axis pedicle screws, acquire the narrowest reconstructed section of the PIC, and precisely measure its width, thus representing the foremost CT assessment method for preoperative evaluation of axis pedicle screw placement [7,8]. However, this method is complex to operate and requires specific software support, and it also entails some degree of subjectivity and error. There is a pressing need for more precise and objective evaluation methods.

Recent years, the rapid advancements in artificial intelligence (AI) technology have provided new approaches for tackling this issue [9–11]. Image processing and deep learning techniques have made remarkable strides in medical imaging, furnishing surgeons with satisfactory image analysis tools [12–14]. Deep learning models have been widely applied in the field of spine and have shown significant progress with great potential [15–19]. In light of the prevailing challenges and opportunities, we have undertaken the development of an AI-powered model dedicated to analyzing cervical spine CT images for precise C2 pedicle screw placement feasibility. Our study aims to introduce advanced methodologies to spinal surgery, offering surgeons a decision support tool to enhance surgical outcomes and safeguard patient welfare.

## Materials and methods

The schematic structure of a deep learning model, named C2-Net, construction and evaluation was shown in Fig 1. The model consisted of two modules: the image segmentation module, which separated the C2 pedicle from the subject's cervical spine CT images, and the probability assessment module, which displayed the feasibility of placing pedicle screws.

## Study samples and data acquisition

This study was implemented at a tertiary teaching hospital in southern China. Ethical approval was obtained from the Medical Research Ethics Committee of the hospital, and written informed consent was waived due to the retrospective nature of the study. Data were accessed for research purposes from 04/12/2023–03/12/2024.

Model training data was collected from patients who underwent cervical spine CT or CTA examinations at a tertiary hospital from 01/01/2017–31/12/2022. The inclusion criteria encompassed subjects with well-defined clinical diagnoses unrelated to vertebral artery anomalies, who had undergone routine 2 mm thin-slice CT scans and 0.4–1.0 mm thin-slice head and neck CTA scans. Conversely, subjects with rheumatoid arthritis, ankylosing spondylitis, spinal metastases, congenital cervical fusion (Klippel-Feil syndrome, e.g.,), or a history of prior head or cervical spine surgery were excluded from the study.

DICOM (Digital Imaging and Communications in Medicine) derived from spinal CT scans of the participants was systematically gathered. The imaging protocol was standardized using a high-resolution 256-slice spiral CT scanner (iCT 256, Philips Healthcare, Amsterdam, Netherlands). Each scan was acquired in spiral mode with the following settings: a layer thickness of 2 mm, a tube voltage of 100 kV, a tube current of 340 mA, a window width of 2000, and a window level of 800. Participants were selected based on predefined inclusion and exclusion criteria to ensure homogeneity and relevance of the data for the intended study objectives.

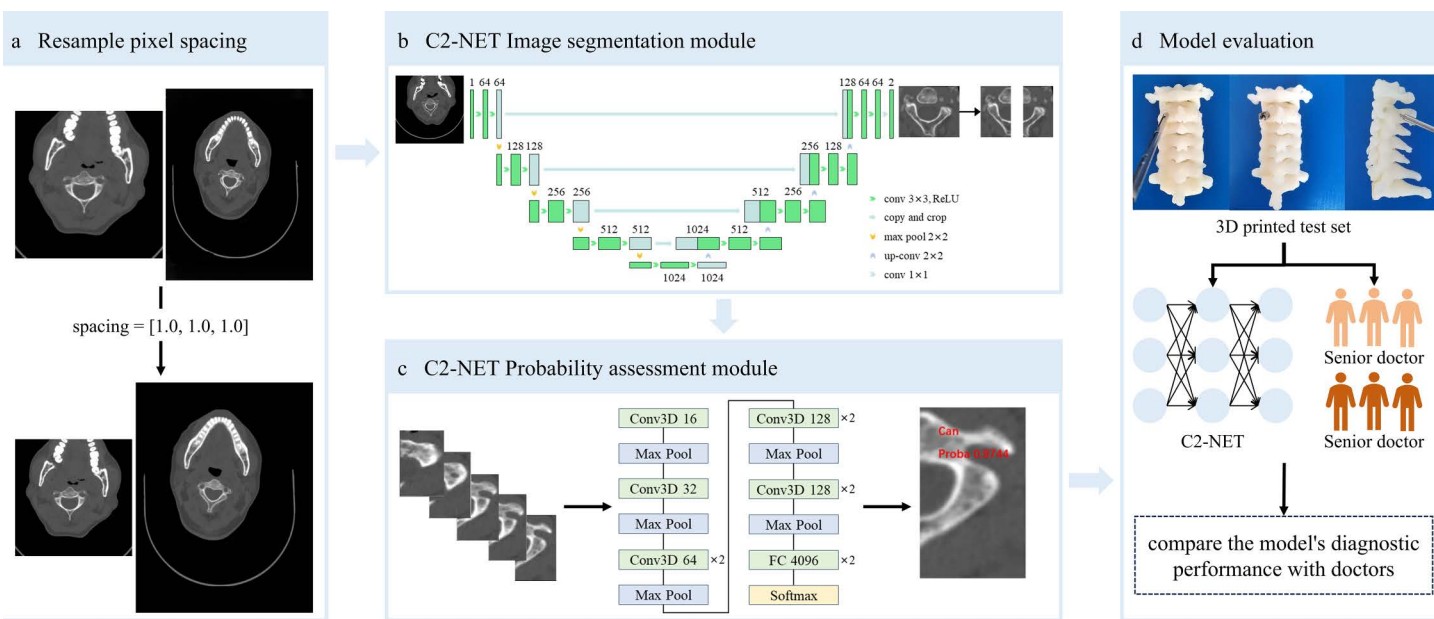

**Fig 1. Schematic structure of the C2-net model construction and evaluation. a** shows the resampling of CT images to standardize pixel spacing across different subject data. **b** depicts the image segmentation module of the C2-Net model, which is used to segment the C2 pedicle regions from CT images. **c** illustrates the probability assessment module of the C2-Net model, which is used to evaluate the feasibility of placing screws in the C2 pedicles. The output of the module is the probability of successful and failed placement. **d** presents the evaluation results of the model, comparing the predicted results with the 3D-printed ground truth and the judgments of surgeons with different levels of experience.

## Training and validation set

In the realm of clinical practice, preoperative evaluations for the placement of C2 pedicle screws are methodically conducted on an individual basis for the left and right pedicles. This necessitates the bifurcation of the C2 pedicle into two distinct segments: left and right, each treated as a separate entity. For analytical clarity, these pedicles are categorized into two risk groups—high-risk and low-risk—based on the minimal pediculoisthmic component diameter (MPD), as measured using the RadiAnt software. The volumetric rendering (VR) and multiplanar reconstruction (MPR) functions are employed to reconstruct the C2 pedicle, achieving a linear measurement precision of 0.01 mm. Labeling was based on our previous CTA-based study in which cortical breach on 3D-printed models served as the reference standard. ROC analysis identified 4.78 mm as the optimal cutoff using the Youden index; pedicles with MPD ≥ 4.78 mm were labeled as low-risk and those < 4.78 mm as high-risk (Fig 2) [7]. To prevent data leakage, splitting was performed at the patient level; both left and right pedicles from the same patient were always assigned to the same subset. Patients were randomly divided into training and validation cohorts at an 8:2 ratio, repeated ten times. Additionally, a sensitivity analysis was performed using an alternative cutoff of 4.30 mm, which was selected as a clinically pragmatic reference threshold based on common empirical considerations of the screw-to-pedicle size relationship, while the labels of the external test set remained unchanged.

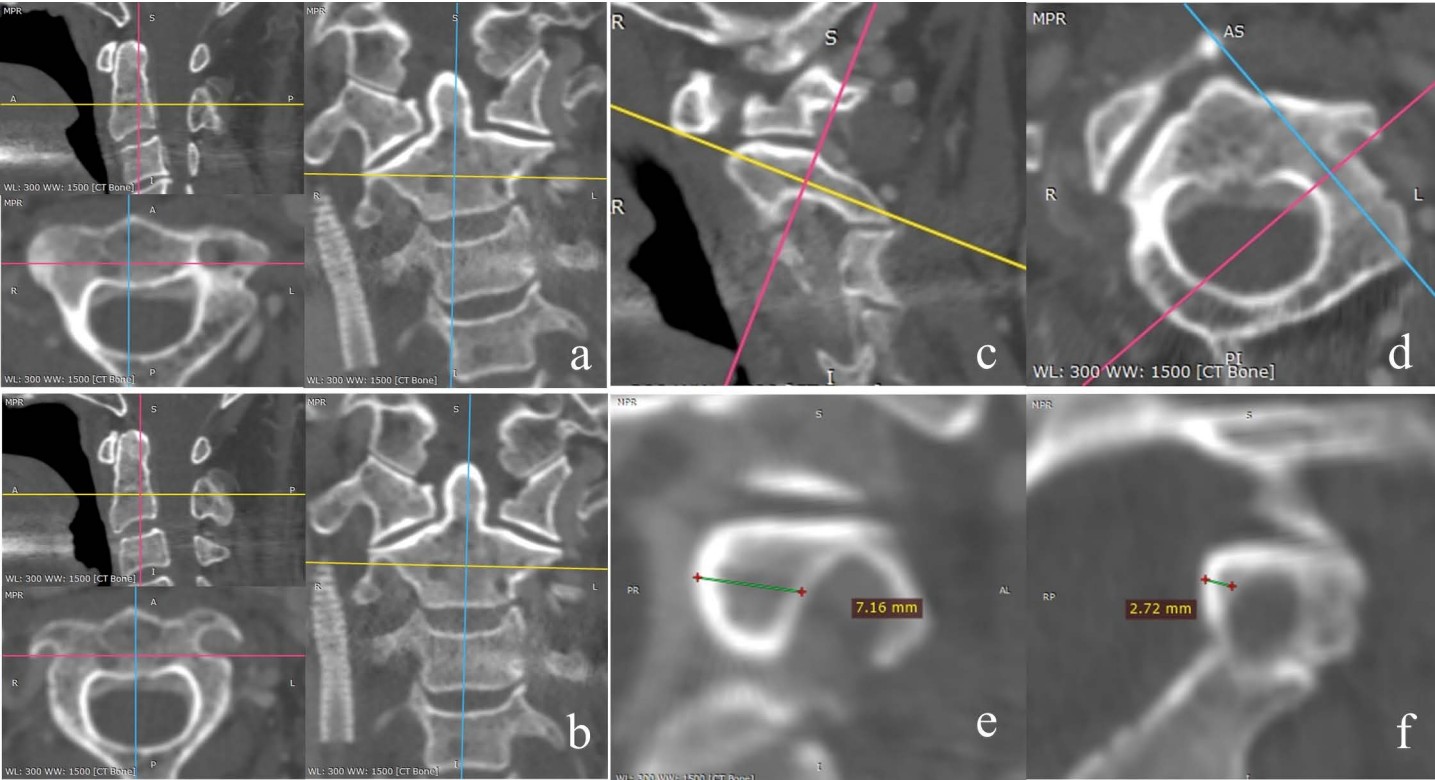

**Fig 2. CT-MPR operation procedure. a**: Utilize the "Multi-Planar Reconstruction (MPR) function" of the RadiAnt DICOM Viewer software; **b**: Employ the MPR function to correct potential skeletal deformities or improper positioning during patient scans to the standard transverse and standard sagittal planes; **c**: Reconstruct a tilted transverse plane along the longitudinal axis of the vertebral arch root notch in the standard sagittal position (yellow line); **d**: Reconstruct an oblique coronal plane perpendicular to the longitudinal axis of the vertebral arch root notch in the tilted transverse plane (red line), ensuring that the oblique coronal section is simultaneously perpendicular to the axis (blue line) and sagittal plane (yellow line) of the root arch, and measure the marrow cavity width (MPD) at the narrowest part of the vertebral arch root notch on the oblique coronal section; **e**: Classify cases with MPD ≥ 4.78 mm into the low-risk group; **f**: Classify cases with MPD < 4.78 mm into the high-risk group.

## 3D printed test set

The confirmation of C2 pedicle screw placement feasibility was determined through simulated surgery on 3D-printed cervical spine models. DICOM data were imported into Mimics software (v23.0; Materialise, Belgium) for 3D bone model reconstruction of C2 vertebrae and exported as STL files. Solid bone models were then printed at a 1:1 scale using a Stratasys J850 3D printer with BoneMatrix RGD516 material, which provides high geometric fidelity and sufficient rigidity for trajectory and cortical breach assessment, although it does not fully replicate the biomechanical properties of living bone. This 3D printing process was supported by the Medical 3D Printing Center of Tengwei Technology.

Using a 3.5mm C2 pedicle screw to manually place screws into the 3D printed model [20]. The ground truth for assessing pedicle wall rupture involves direct visualization and CT scanning (Fig 3). If a pedicle wall rupture was detected, the

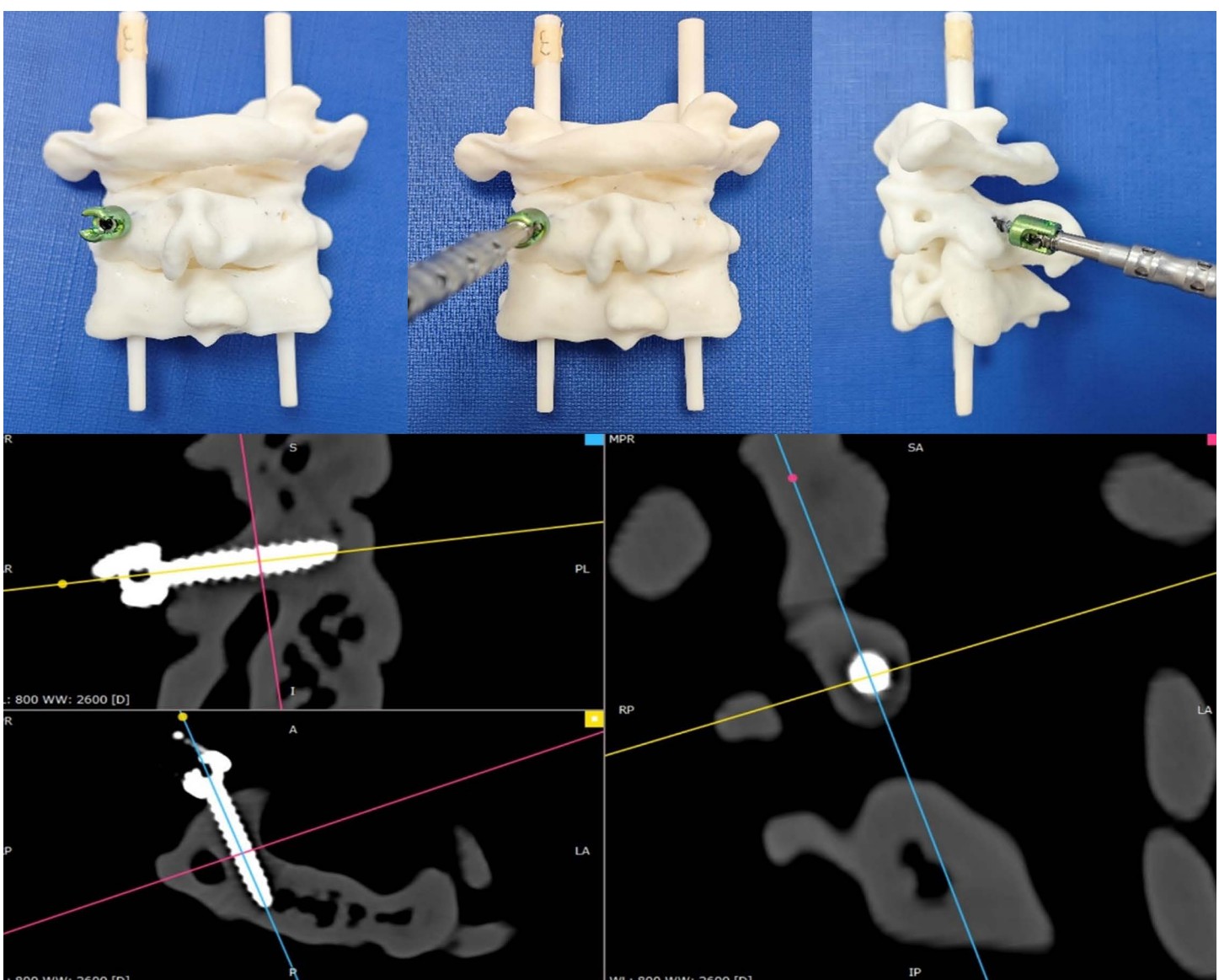

**Fig 3. 3D-printed model and CT assessment of C2 pedicle screw insertion.** C2 pedicle screw placement in a 3D-printed bone model, followed by CT multiplanar reconstruction to evaluate screw trajectory and cortical integrity.

C2 pedicle was placed into the high-risk group of the test set; conversely, if no rupture was observed, it was classified into the low-risk group.

## Image processing

To prepare data for analysis, DICOM images in the training, validation, and test sets were converted to JPEG format. Resampling techniques were used to standardize pixel spacing across CT datasets, by standardizing the sampling interval to 1.0 across all axes, the processed image exhibits uniform physical dimensions along all axes, ensuring comparability of C2 pedicle among subjects. Initially, the ratio between original and desired pixel spacing was calculated and applied to interpolate images, maintaining spatial resolution. Image standardization techniques addressed CT image discrepancies, ensuring consistency. Five CT images per C2 pedicle were selected based on pedicle isthmus narrowing, ranging from wide to narrow, as input data. All CT images were tri-channel grey images (R = G = B). During standardization, mean values were subtracted from R, G, and B values and divided by standard deviation, yielding standardized outputs. Mean and standard deviation were calculated from R, G, and B values within the training set.

## Deep learning model construction

C2-Net is an automated pipeline that includes an image segmentation module for delineating C2 pedicles in CT images and a screw placement probability assessment module for evaluating the feasibility of screw placement. These modules consist of standard deep learning components, including convolutional layers, pooling layers, nonlinear activation functions, dropout, and fully connected layers, enabling automated feature extraction and end-to-end learning.

The segmentation module employed a U-Net architecture with a five-level encoder–decoder structure and skip connections. The number of feature channels in the encoder progressively increased from 64 to 512, and the decoder symmetrically restored spatial resolution by upsampling and feature concatenation. All convolutional layers used $3 \times 3$ kernels with ReLU activation. This module enabled accurate localization and isolation of the C2 pedicles from surrounding anatomical structures.

Following segmentation, each C2 pedicle was separated into left and right components, which were treated as independent analytical units. For each pedicle, five consecutive axial CT slices centered on the narrowest pedicle region were extracted as input for feasibility assessment.

The screw placement feasibility assessment module was implemented using a modified C3D network comprising four 3D convolutional blocks with channel sizes of 16, 32, 64, and 128, respectively [21,22]. Each block consisted of $3 \times 3 \times 3$ convolutions with padding, ReLU activation, and max-pooling. The extracted features were subsequently processed through three fully connected layers (4096, 1024, and 2 neurons), with the final layer outputting the probability of screw placement feasibility (low-risk vs high-risk). Dropout (rate = 0.5) was applied to reduce overfitting.

Model training was performed using the Adam optimizer with an initial learning rate of $1 \times 10^{-4}$ and a batch size of 4. A cosine annealing learning rate scheduler was applied, and training was conducted for up to 200 epochs with early stopping if the validation loss did not improve for 10 consecutive epochs.

The overall training workflow is illustrated in Fig 1, and the final model parameters were selected based on the best performance on the validation set.

## Screw placement probability assessment

The model categorizes split pedicles into high-risk group and low-risk group and generates a GIF illustrating the probability of C2 pedicle screw placement.

After analyzing the input data, the model outputs raw scores, known as logits, for each category. Subsequently, the Softmax function is applied to calculate the predicted probabilities, transforming the logits into a probability distribution. For the logits vector $[z_1, z_2,..., z_n]$ of each input sample, the Softmax function is computed using the formula:

$$\text{Softmax}(z_i) = \frac{e^{z_i}}{\sum_{j=1}^{n} e^{z_j}}$$

$e^{z_i}$ represents the exponentiation of logit $z_i$, and the denominator is the sum of the exponentiations of all logits. This ensures that the sum of probabilities for all categories equals 1. Through this approach, the model not only classifies split pedicles into those high-risk and low-risk for screw placement but also provides the probability of screw placement feasibility. This aids in assessing the model's confidence and performance.

### The screw placement probability assessment module training and validation

After segmentation, each of the split C2 pedicles, which includes five segmented CT images, was fed into the screw placement probability assessment module to evaluate the overall risk and predict the feasibility of screw placement. The screw placement probability assessment module underwent training for 200 epochs. In each epoch, the model received the split C2 pedicle's images and labels from the training set, resulting in the generation of model parameters. Model parameters were saved every 10 epochs for validation using the validation set. Parameters demonstrating optimal diagnostic efficacy were selected as the final model parameters and subjected to further validation using the external test set.

### Model evaluation

The model's performance was externally validated through a 3D printed test set. Additionally, the 3D printed test set was evaluated by surgeons of varying experience levels to compare the model's diagnostic performance with that of physicians, thereby validating the model's credibility. (Senior spine surgeons: Dr. Zhang and Dr. Wang; Junior spine surgeons: Dr. Lin and Dr. Liang)

Senior spine surgeons in this study had over 10 years of experience and held senior professional titles, while junior spine surgeons had less than 5 years of experience and held intermediate or lower professional titles.

### Model attention

The deep learning visualization method was used to generate an attention map to show the areas identified by C2-Net that attracted the most attention of the model. A cut-off value of 0.5 was used to preserve the high response of the attention area.

### Statistical analysis

Continuous variables are presented as means ± standard deviations, while discrete variables are expressed as frequencies and percentages. Evaluation of C2-Net's classification performance utilized metrics including accuracy, sensitivity, specificity, area under the receiver operating characteristic curve (AUC). These metrics were reported as percentages with corresponding 95% confidence intervals (CIs). All deep learning methods were performed using the PyTorch toolkit and Python 3.7 (Python Software Foundation, www.python.org).

## Results

### Data acquisition and dataset splitting

A flowchart of subject data retrieval and the study is shown in Fig 4, and the demographics of the two cohorts of enrolled subjects are shown in Table 1. A total of 490 split C2 pedicles were included, 245 were categorized as high-risk, and 245 were deemed as low-risk. Among these, 440 pedicles were proportionally assigned to the training and validation sets, while the remaining 50 cases were the external test set, with a total of 2450 C2 pedicles CT images. We used MPR to measure the narrowest diameter of the C2 pedicle isthmus. The average narrowest width of the left C2 pedicle isthmus was 5.82 ± 1.61 mm, and the right was 5.55 ± 1.44 mm. The left PIC measurement was significantly greater than the right.

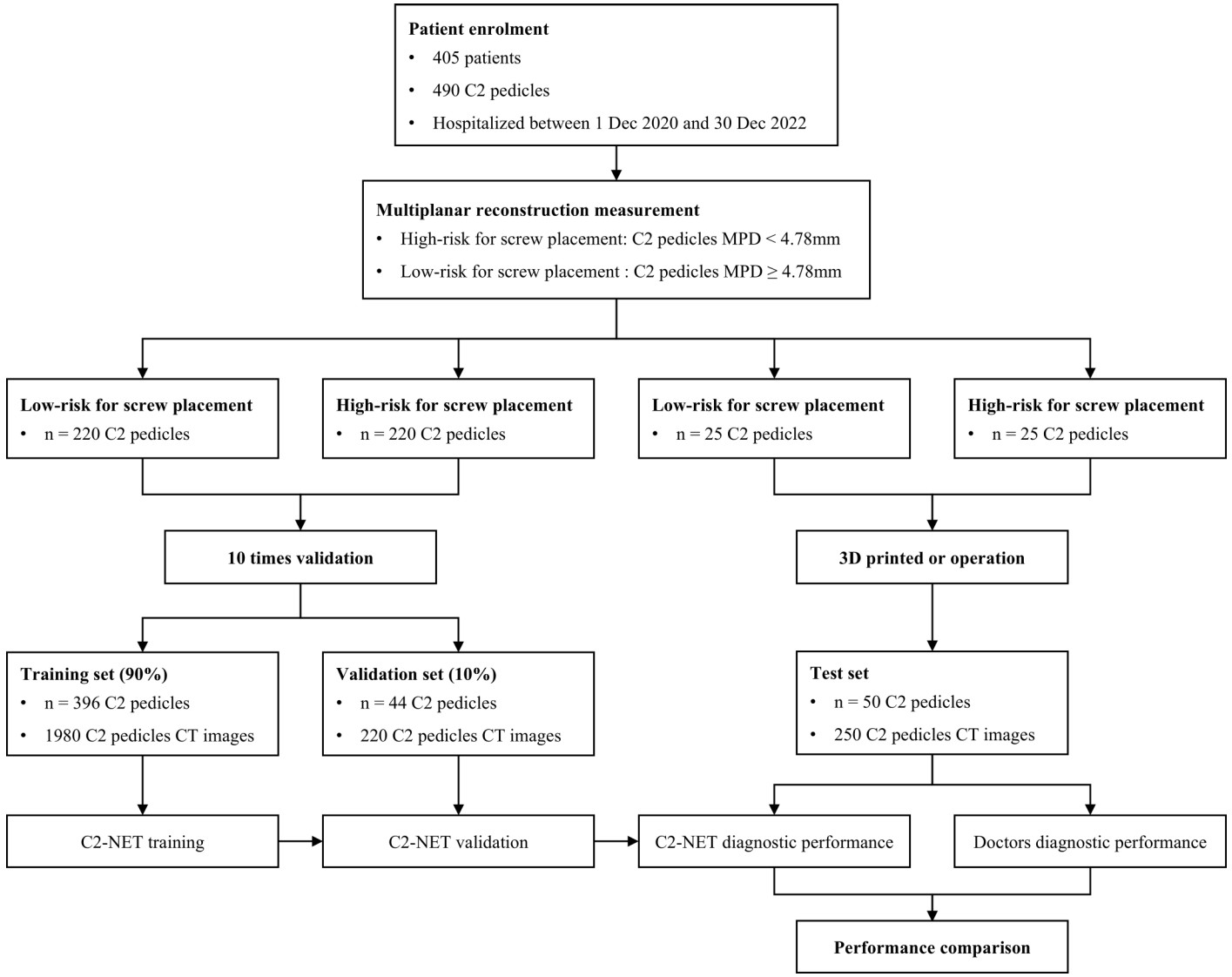

**Fig 4. Flowchart of subject data retrieval and the study.** 405 patients (490 C2 pedicles) were enrolled and stratified by pedicle diameter into low- and high-risk groups for screw placement. Both groups were included in the 10-fold cross-validation for C2-NET training and validation, while an independent test set was used to compare the AI model's performance with that of doctors.

### The image segmentation module performance

The image segmentation module of C2-Net demonstrates high performance, accurately segmenting the C2 pedicle from CT images. In the validation set for image segmentation, it achieved a dice coefficient of 0.959.

### Model diagnostic efficacy

The model's average accuracy in determining whether C2 pedicle screw placement was feasible in the validation set was 91.4%, with an average AUC of 0.94 (95% CI, 0.91 to 0.97) and average sensitivity and specificity of 0.93 and 0.90, respectively. In the 3D printed test set, the model's average accuracy was 89.4%, the average AUC was 0.94 (95% CI, 0.91 to 0.96), and the average sensitivity and specificity were 0.90 and 0.89, respectively (Fig 5). No statistically significant

**Table 1. Summary of clinical and imaging features of study subjects and C2 minimum pediculoisthmic component diameter.**

| Characteristics | Development dataset | Test dataset |
|---|---|---|
| N of patients | 320 | 37 |
| N of PED | 440 | 50 |
| Sex | | |
| Female | 154 (51.98%) | 23(62.16%) |
| Male | 166 (48.02%) | 14(37.84%) |
| Mean age±SD (y) | 54±14 | 61±15 |
| Width | | |
| Left | 5.63±1.66 | 5.58±2.14 |
| Right | 5.35±1.43 | 4.72±2.38 |
| Feasibility | | |
| Yes | 220 (50%) | 25(50%) |
| No | 220 (50%) | 25(50%) |

PED: pedicle; SD: Standard Deviation

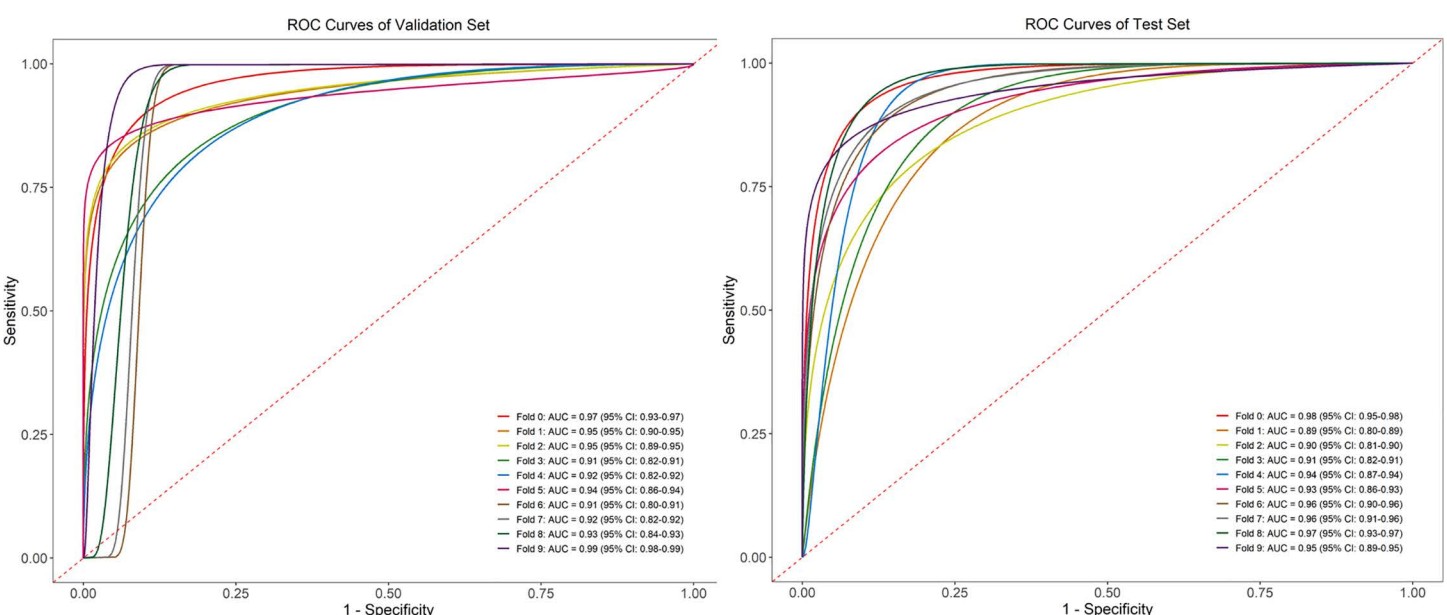

**Fig 5. ROC curve of the validation, test set.** ROC curves for the C2-NET model, showing performance in the 10-fold cross-validation (left) and independent test set (right). Each curve corresponds to a validation fold, with AUC values indicating strong and consistent diagnostic accuracy.

differences were observed between the validation and test sets across these performance metrics. Screw placement probability was displayed as a GIF by combing the input five images of the C2 pedicle, which was illustrated in the Fig 1.

## Comparison with clinical surgeons

Compared to predictions made by junior surgeons, the C2-Net model showed a slight advantage in assessing the feasibility of C2 pedicle screw placement, achieving an accuracy rate and AUC of 89.4% versus 88.0% ($P > 0.05$), and 0.94 (95%

CI, 0.91 to 0.96)versus 0.88 ($P > 0.05$), respectively. However, when contrasted with senior surgeons' assessments, the human experts outperformed the C2-Net model, boasting an accuracy rate and AUC of 96.0% versus 89.4% ($P > 0.05$), and 0.98 versus 0.94 (95% CI, 0.91 to 0.96, $P > 0.05$), respectively (Fig 6). Moreover, the senior surgeon group demonstrated significantly higher accuracy and sensitivity than the junior surgeon group ($P < 0.05$).

### Visual interpretation of deep-learning internal decision making

Attention area in the CT images detected by C2-Net are shown in Fig 7. Regardless of whether the validation or test set was used, the attention maps showed that C2-Net was able to detect pedicle isthmus and label them as highly responsive areas, indicating that the model can adequately learn the features of C2 pedicle CT images and respond appropriately.

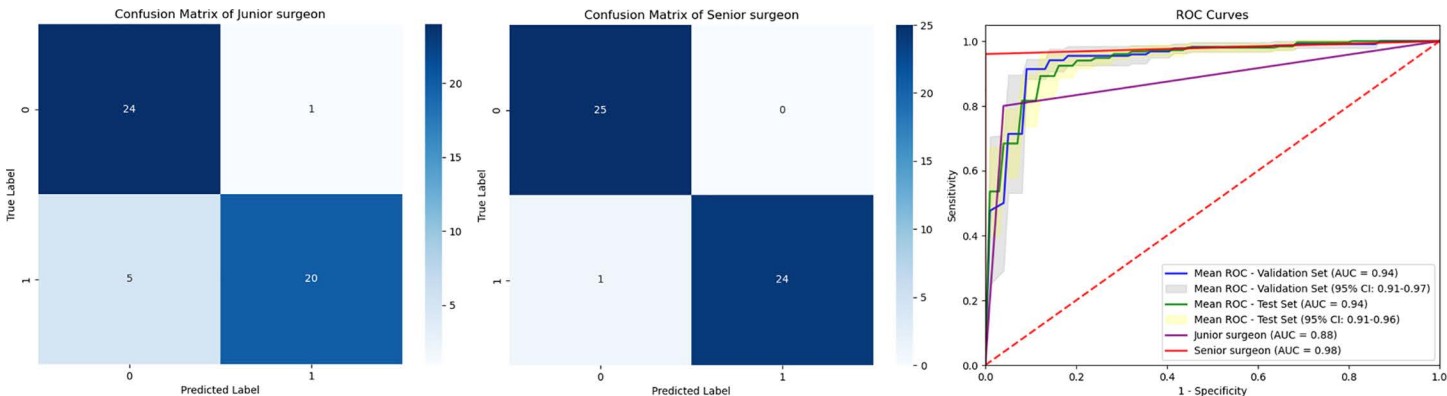

**Fig 6. Performance comparison of C2-net model with junior and senior surgeons in C2 pedicle screw placement assessment.** Confusion matrices illustrate the diagnostic outcomes of junior and senior surgeons, while the ROC curves compare their performance to the C2-NET model, demonstrating the model's superior accuracy in identifying high-risk C2 pedicles.

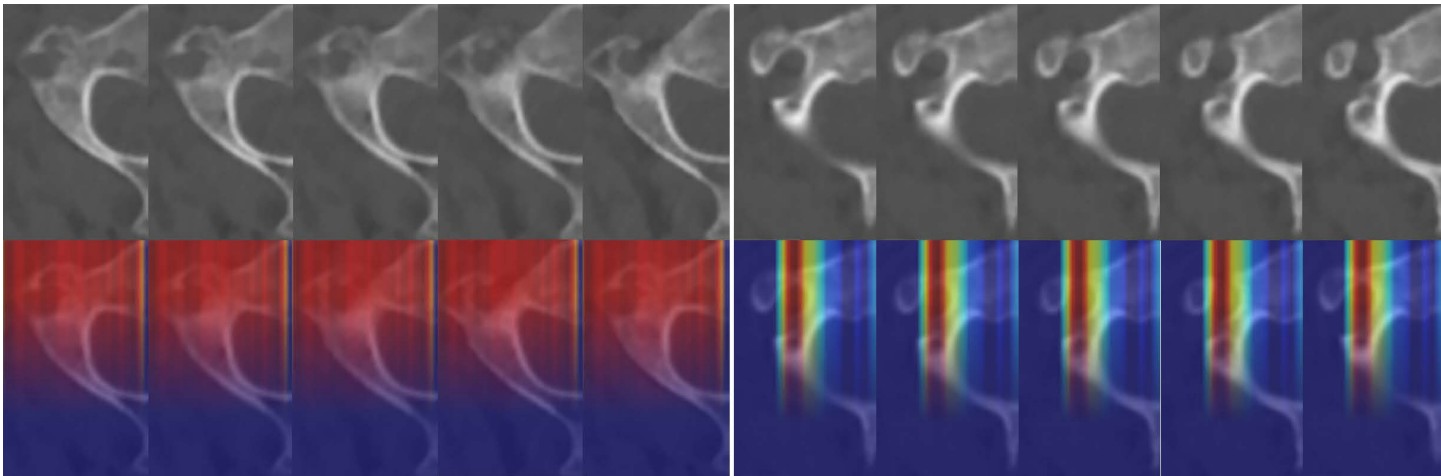

**Fig 7. Attention area in the CT images detected by C2-Net.** Original CT images (top) and corresponding model attention maps (bottom) show the regions of interest the C2-NET model focuses on to evaluate C2 pedicle screw placement risk, with color intensity reflecting the degree of attention.

## Discussion

This study introduced C2-Net model, a deep learning model combining image segmentation and probability assessment functionalities, offering an end-to-end solution for pedicle screw placement surgical planning and improving both accuracy and efficiency.. Model performance was evaluated by performing pedicle screw placement surgery on 3D printed C2 pedicle models. Through this deep model, we were able to integrate the feasibility assessment of placing pedicle screws in the C2 PIC into a complete, automated workflow, thus improving the accuracy and efficiency of the evaluation. Similar AI-based models have been developed for lumbar or thoracic pedicle screw planning, demonstrating high accuracy and clinical feasibility, highlighting the growing potential of deep learning in spinal surgical planning [23–25].

Various CT- or CTA-based techniques have been employed to assess the morphometric characteristics of the C2 PIC for determining the safety and feasibility of C2 pedicle screw or transarticular screw placement [26–28].These methods include transverse C2 pedicle width, defining HRVA, and oblique CT scan reconstructions. Reported screw misplacement rates vary from 5% to 41%, reflecting the influence of measurement techniques, surgical experience, and anatomical variability [29]. In contrast, C2-Net uses a 3D convolutional (C3D) architecture that analyzes volumetric image sequences to capture richer anatomical context, outperforming traditional 2D or slice-based analyses [30,31]. This design ensures higher diagnostic performance, reproducibility, and robustness across different imaging sources. However, because this study focused solely on pedicle morphometry, vascular variants of the vertebral artery and bone quality were not incorporated into the current model. Although the model's performance did not differ significantly from that of either the senior or junior surgeon groups, this may be due to the limited sample size. Nevertheless, the model demonstrated a performance pattern more consistent with senior surgeons across multiple metrics, including accuracy, sensitivity, and specificity, and showed numerically higher but not statistically significant performance compared with junior surgeons. Moreover, a significant difference in both sensitivity and accuracy was found between the senior and junior surgeon groups, suggesting that surgical experience plays an important role in maintaining diagnostic consistency.

The definition of a "narrow C2 pedicle" remains controversial, as different CT measurement techniques yield variable pedicle diameter values, leading to inconsistent criteria for determining screw placement feasibility. Maki et al reported that pedicles with a medullary canal width ≤ 4 mm were unsuitable for safe C2 pedicle screw insertion [32]. Similarly, Marques et al used the MPR function of OsiriX and proposed that pedicle widths of at least 5.5 mm and 6.0 mm are required for 3.5 mm and 4.0 mm screws, respectively [33]. In this study, based on ROC analysis of cortical breach observed in 3D-printed models, the Youden index identified 4.78 mm as the optimal MPD cutoff) [7]. This threshold was validated in our earlier work, where simulated insertion of a 3.5 mm screw frequently resulted in cortical breach when the pedicle width was less than 4.78 mm. Using this standardized cutoff allowed C2-Net to achieve diagnostic performance comparable to that of senior surgeons, supporting its potential as an objective and reproducible tool for preoperative risk assessment. In contrast, When a more conservative alternative cutoff based on the commonly used 80% screw-to-pedicle ratio was applied for sensitivity analysis, model performance decreased, supporting the robustness of the ROC-derived 4.78-mm threshold(Supplementary S1 Table). However, this threshold was derived from single-center data primarily involving an Asian population and may not be directly generalizable to other ethnic groups with different anatomical morphologies. In addition, subtle differences in operator measurement techniques, CT resolution, and segmentation precision may introduce variability near this boundary. Therefore, the 4.78-mm cutoff should be regarded as a statistical reference rather than an absolute safety limit. Future research should explore adaptive or patient-specific thresholding strategies or develop a continuous risk-scoring system to more flexibly represent the probability of cortical breach..

Due to the limited availability and high cost of cadaveric specimens, BoneMatrix 3D-printed bone models were used to simulate the C2 pedicle screw placement process. These models replicate cortical thickness and biomechanical characteristics of the C2 pedicle, allowing a controlled and reproducible testing environment while overcoming sample size limitations. Although the X-ray absorption differences between the model cortex and marrow cannot be completely distinguished, this limitation has minimal impact on the evaluation of screw trajectory or cortical breach detection.

Clinically, C2-Net demonstrates strong potential for application. It can provide less experienced surgeons with rapid, standardized, and reproducible preoperative assessments. By optimizing workflow, it can also save time and labor costs, improve resource utilization, and enhance reproducibility across institutions. Ultimately, such automation can promote safer and more individualized treatment strategies, potentially improving patient outcomes in spine surgery.

While the overall accuracy and AUC of the model demonstrated strong performance comparable to senior surgeons, detailed analysis of false-positive and false-negative predictions provides further insight into its clinical reliability. False positives—cases in which trajectories are incorrectly predicted as high-risk despite being safe—may lead to unnecessary alterations in fixation strategies or avoidance of optimal screw paths, thereby increasing operative time and surgical complexity. Conversely, false negatives—cases predicted as safe when they are actually high-risk—pose substantial clinical danger, potentially resulting in vertebral artery, spinal cord, or nerve root injury. In this study, the senior surgeons demonstrated the lowest false-positive rate (4%), significantly lower than that of the junior group (20%, $P = 0.037$). The AI model's false-positive rate showed no significant difference compared with either surgeon group, indicating its comparable capability in minimizing false risk assessment. False-negative rates were generally low and did not differ significantly among the three groups. Future optimization of the model will focus on improving the recognition of borderline cases by increasing the weighting of high-risk samples during training and integrating human-in-the-loop validation, allowing surgeons to review model outputs through visualization and attention maps before making final decisions.

This study still has several limitations. The training dataset primarily consisted of Asian patients, requiring external validation across diverse populations and multi-center settings to ensure generalizability. Direct in vivo validation was not performed, and 3D-printed models cannot fully replicate intraoperative environments. In addition, the external test cohort of 50 pedicles limits statistical power and may underestimate performance variability. Therefore, future work will include prospective clinical validation using CTA-based blind prediction compared with postoperative CT outcomes, expansion of external test cohorts, and integration of C2-Net into navigation or surgical planning systems to build a comprehensive AI-assisted surgical planning platform.

## Conclusion

The C2-Net model represents a significant advancement in the preoperative assessment of C2 pedicle screw placement feasibility. By combining deep learning techniques for image segmentation and probability assessment, C2-Net offers an automated, efficient, and accurate solution for surgical planning. The model's performance is comparable to that of experienced surgeons, while maintaining time-efficiency and reducing subjective errors. This advancement promises to enrich clinical practice, optimize patient care, and ultimately contribute to improved surgical outcomes and patient safety.

## Supporting information

**S1 Table. Sensitivity analysis of model performance under an alternative cutoff definition (All results are based on the same 3D-printed test set).**
(DOCX)

## Author contributions

**Conceptualization:** Fengshi Jing, Li Zhang.

**Data curation:** Junhao Bao, Hao Ren, Zhaoquan Liang, Qiang Xiao.

**Formal analysis:** Junhao Bao, Wei Wang.

**Investigation:** Yeyang Wang, Li Zhang.

**Supervision:** Weibin Cheng.

**Validation:** Wei Wang, Yuelin Wu.

**Writing – original draft:** Junhao Bao, Wei Wang.

**Writing – review & editing:** Weibin Cheng, Li Zhang.

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
