## [Decision Letter · Decision Letter 0]

8 Dec 2025

Dear Dr. Zhang,

Thank you for submitting your manuscript to PLOS ONE. After careful consideration, we feel that it has merit but does not fully meet PLOS ONE’s publication criteria as it currently stands. Therefore, we invite you to submit a revised version of the manuscript that addresses the points raised during the review process.

We look forward to receiving your revised manuscript.

Kind regards,

Dean Chou, MD

Academic Editor

PLOS One

Journal Requirements:

2. Please note that PLOS One has specific guidelines on code sharing for submissions in which author-generated code underpins the findings in the manuscript. In these cases, all author-generated code must be made available without restrictions upon publication of the work. Please review our guidelines at https://journals.plos.org/plosone/s/materials-and-software-sharing#loc-sharing-code and ensure that your code is shared in a way that follows best practice and facilitates reproducibility and reuse.

“This work was supported by the 3D-printing research project of Guangdong Second Provincial General Hospital (3D-A2020006) and the Guangzhou Science and Technology Programme (2024A03J1062, 2024A03J074, 2023A03J0286, and 2024A03J0927).”

“This work was supported by the 3D-printing research project of Guangdong Second Provincial General Hospital (3D-A2020006) and the Guangzhou Science and Technology Programme (2024A03J1062, 2024A03J074, 2023A03J0286, and 2024A03J0927).”

“This work was supported by the 3D-printing research project of Guangdong Second Provincial General Hospital (3D-A2020006) and the Guangzhou Science and Technology Programme (2024A03J1062, 2024A03J074, 2023A03J0286, and 2024A03J0927).”

6. In the online submission form you indicate that your data is not available for proprietary reasons and have provided a contact point for accessing this data. Please note that your current contact point is a co-author on this manuscript. According to our Data Policy, the contact point must not be an author on the manuscript and must be an institutional contact, ideally not an individual. Please revise your data statement to a non-author institutional point of contact, such as a data access or ethics committee, and send this to us via return email. Please also include contact information for the third party organization, and please include the full citation of where the data can be found.

7. Please include a separate caption for each figure in your manuscript.

Reviewers' comments:

Reviewer's Responses to Questions

**Comments to the Author**

1. Is the manuscript technically sound, and do the data support the conclusions?

Reviewer #1: Yes

Reviewer #2: Yes

Reviewer #3: Partly

2. Has the statistical analysis been performed appropriately and rigorously?

Reviewer #1: Yes

Reviewer #2: Yes

Reviewer #3: Yes

3. Have the authors made all data underlying the findings in their manuscript fully available?

Reviewer #1: Yes

Reviewer #2: Yes

Reviewer #3: No

4. Is the manuscript presented in an intelligible fashion and written in standard English?

Reviewer #1: Yes

Reviewer #2: Yes

Reviewer #3: Yes

Reviewer #1: This manuscript presents the development and validation of a deep learning–based workflow (C2-Net) for preoperative evaluation of C2 pedicle screw placement feasibility using CT images. The authors integrated an automated segmentation model and a classification module, and validated performance against a 3D-printed model–based ground truth and surgeons of different experience levels. However, several areas require clarification and expansion to strengthen the article.

1.The 4.78-mm MPD cutoff requires stronger justification

The chosen cutoff results in an almost perfectly balanced distribution (245 high-risk vs. 245 low-risk pedicles). This raises concerns about whether the threshold reflects clinical reality or merely statistical optimization.

Please provide:

1)A detailed explanation of how 4.78 mm was determined.

2)Whether a ROC-based Youden index was used in previous work.

3)Sensitivity analyses using alternative thresholds (e.g., 4.0 mm, 5.0 mm).

2.Deep learning module descriptions are insufficient for reproducibility. Detailed reporting of model architectures and training are required. Like, complete U-Net configuration (encoder/decoder depth, filters, image size); Full C3D architecture (kernel sizes, number of channels, FC layer dimensions); Optimization details: learning rate, optimizer type, batch size, dropout, epoch count, early stopping criteria; Data augmentation strategies, if any; Measures taken to prevent data leakage (e.g., patient-level separation).

Reviewer #2: Abnormalities in the C2 pedicle root led to the failure of screw placement. The main issues included the overly thin diameter of the pedicle root, variations in the vertebral artery, and the bone condition of the pedicle root, such as osteoporosis, bone destruction or hyperplasia. However, this study excluded variations in the vertebral artery, as it solely relied on the diameter of the pedicle root as the criterion. Moreover, it performed screw placement on 3D printed specimens, which did not match the bone condition of the human body. We know that this varies from person to person. Ultimately, the diameter of the pedicle root was only used as one of the reference reasons. Therefore, I believe this research should continue. It should be verified in surgical patients to determine the success of this model. For example, using the CT imaging data of patients who have undergone surgery for a double-blind assessment, and then comparing it with the CT obtained during postoperative follow-up to evaluate it, this would be more convincing.

Reviewer #3: This is a unique study that uses AI to determine the feasibility of C2 pedicle screw insertion compared to junior and senior surgeons. The statistical descriptions are technically sound but there are several concerns that need to be addressed:

• It appears that the MPD cutoff 4.78 mm is central to the whole study. The authors need to provide further discussion on how this value was picked and provide analyses of different MPD cutoffs

• Further clinical context on the application of their tool needs to be provided. Brief discussion of how C2-Net could integrate into navigation/planning systems and handle patients with vertebral artery anomalies.

• The claim that the model outperforms junior residents is not supported in the results (P>0.05). Perhaps this is a typo in the results?

• The external test of 50 patients is small and this needs to be discussed in their limitations

• The authors need to explicitly discuss data leakage risks of the left and right pedicles appearing in different splits

**Do you want your identity to be public for this peer review?** For information about this choice, including consent withdrawal, please see our Privacy Policy

Reviewer #1: No

Reviewer #2: No

Reviewer #3: No

---

## [Author Response · Author response to Decision Letter 1]

19 Jan 2026

Dear Dr. Chou and Reviewers,

We sincerely thank the Academic Editor and all reviewers for their careful evaluation and constructive comments. We have revised the manuscript thoroughly in accordance with the suggestions. All modifications are highlighted in the tracked-changes version. Below we provide a detailed, point-by-point response. Line numbers refer to the revised manuscript with track changes.

Response to Journal Requirements

1. Compliance with PLOS ONE style

We have revised the manuscript to fully comply with PLOS ONE formatting and file-naming requirements, including figure captions, references, and structure.

2. Code sharing

All author-generated code has been made publicly available without restrictions at: https://github.com/Ingram66/C2-NET

3–5. Funding statement

Funding information has been removed from the Acknowledgments section and retained only in the Funding Statement. The grant number has been corrected from 2024A03J074 → 2024A03J1074.

6. Data statement

In accordance with PLOS policy, the contact point has been changed to a non-author institutional office: The Medical Ethics Committee of Guangdong Second Provincial General Hospital (Email: gd2hllwyh@163.com).

This has been updated in the Data Statement (Lines 437–439).

7. Figure captions

Separate captions for each figure have now been provided in the manuscript.

Response to Reviewer #1

Comment 1

The 4.78-mm MPD cutoff requires stronger justification… sensitivity analyses using alternative thresholds are recommended.

Response

We appreciate this important comment. We have clarified the origin and role of the 4.78-mm threshold:

1. Derivation of 4.78 mm

The 4.78-mm threshold was derived from our previous independent CTA-based 3D-printed study using ROC analysis and the Youden index, and was not optimized on the current dataset. In the present study, this value was used only for data labeling (MPD ≥4.78 mm as low-risk; <4.78 mm as high-risk) and not as an intraoperative decision rule. Importantly, the labels in the external test set were determined by actual screw placement outcomes on 3D-printed models, rather than by MPD measurements, thereby avoiding circular validation.

Added in Materials and methods--Training and validation set, Lines 109-111

Clarified in Discussion, Lines 362-366

2. Sensitivity analysis

A supplementary analysis was conducted using an alternative 4.30-mm cutoff, selected with reference to the commonly accepted 80% screw–pedicle ratio principle, rather than as a mathematically exact derivation from the 3.5-mm screw diameter. The model was retrained using this threshold, while the external test set and its labels—based on cortical breach observed in 3D-printed screw placement—remained unchanged to avoid data leakage. Under this setting, model performance decreased compared with the ROC-derived 4.78-mm threshold, which supports the robustness and clinical appropriateness of the original cutoff.

Added in Materials and methods--Training and validation set, Lines 115-118

Added in Discussion, Lines 369-372

Results provided in Supplementary Table S1

Comment 2

Deep learning module descriptions are insufficient for reproducibility.

Response

We agree that detailed methodological reporting is essential. Accordingly, we have revised the manuscript to comprehensively describe the model architecture, training strategy, and data handling procedures to facilitate reproducibility.

Added in Materials and methods -- Deep learning model construction, Lines 176-196.

Comment 3

Risk of data leakage between left and right pedicles.

Response

Dataset partitioning was performed strictly at the patient level before any preprocessing or augmentation. All slices and both left/right pedicles from the same patient were assigned exclusively to one subset (training, validation, or testing). Therefore, no image, slice, or anatomical structure from a given patient was ever present across different subsets, effectively preventing data leakage.

Clarified in Materials and methods--Training and validation set, Lines 112-116

Response to Reviewer #2

Comment

The study did not consider vertebral artery anomalies and used 3D-printed models rather than real bone conditions.

Response

We thank the reviewer for this important comment. We acknowledge that the present study did not incorporate vertebral artery anomalies and that the experiments were performed on 3D-printed models rather than real intraoperative bone conditions.

In this study, cases with high-riding vertebral artery or other vascular variations were intentionally excluded, and the proposed model was designed to evaluate pedicle morphology only. We have now expanded the Discussion to clarify that clinical application of the system requires integration with CTA-based vascular assessment and that prospective validation in real surgical patients will be necessary to confirm safety and generalizability.

Added in Discussion, Lines 347-349 and 415-419

Response to Reviewer #3

Comment 1

Need further discussion of the 4.78-mm value and alternative cutoffs.

Response

Addressed as described for Reviewer #1. A sensitivity analysis is now presented (Supplementary Table S1).

Comment 2

Claim that the model outperforms junior residents is not supported (P>0.05).

Response

Thank you for noting this. The statement has been corrected to reflect no statistically significant difference between C2-Net and junior surgeons.

Revised in Discussion, Lines 351-355

Comment 3

External test set of 50 patients is small.

Response

This limitation has been explicitly acknowledged.

Added in Discussion, Lines 418-420

Comment 4

Need explicit discussion of data leakage.

Response

Addressed as above;

Clarified in Materials and methods--Training and validation set, Lines 112-116

Supplementary Material

Cutoff scheme Accuracy Sensitivity Specificity AUC

4.78 mm

(primary, ROC-derived) 0.89 0.9 0.89 0.94

80% screw–pedicle ratio (sensitivity analysis) 0.66 0.72 0.6 0.71

Table S1. Sensitivity analysis of model performance under an alternative cutoff definition (same 3D-printed test set)

We sincerely appreciate the reviewers’ valuable suggestions, which have substantially improved the manuscript. We hope the revised version is now suitable for publication in PLOS ONE.

Sincerely,

Li Zhang

---

## [Editor Report · Decision Letter 1]

22 Jan 2026

Preoperative Evaluation of C2 Pedicle Screw Placement Using a Deep Learning Model: Development and Validation Study

PONE-D-25-56509R1

Dear Dr. Zhang,

We’re pleased to inform you that your manuscript has been judged scientifically suitable for publication and will be formally accepted for publication once it meets all outstanding technical requirements.

Kind regards,

Dean Chou, MD

Academic Editor

PLOS One
---

## [Editor Report · Acceptance letter]

PONE-D-25-56509R1

PLOS One

Dear Dr. Zhang,

I'm pleased to inform you that your manuscript has been deemed suitable for publication in PLOS One. Congratulations! Your manuscript is now being handed over to our production team.

Kind regards,

on behalf of

Dr. Dean Chou

Academic Editor

PLOS One